# DEEP AMORTIZED CLUSTERING

## ABSTRACT

We propose a *deep amortized clustering* (DAC), a neural architecture which learns to cluster datasets efficiently using a few forward passes. DAC implicitly learns what makes a cluster, how to group data points into clusters, and how to count the number of clusters in datasets. DAC is meta-learned using labelled datasets for training, a process distinct from traditional clustering algorithms which usually require hand-specified prior knowledge about cluster shapes/structures. We empirically show, on both synthetic and image data, that DAC can efficiently and accurately cluster new datasets coming from the same distribution used to generate training datasets.

## 1 INTRODUCTION

Clustering is a fundamental task in unsupervised machine learning to group similar data points into multiple clusters. Aside from its usefulness in many downstream tasks, clustering is an important tool for visualising and understanding the underlying structures of datasets, as well as a model for categorisation in cognitive science.

Most clustering algorithms have two basic components - how to define a cluster and how to assign data points to those clusters. The former is usually defined using metrics to measure distances between data points, or using generative models describing the shapes of clusters. The latter, how to assign data points to the clusters, is then typically optimized iteratively w.r.t. objective functions derived based on the cluster definitions. Note that cluster definitions are user defined, and are reflections of the user's prior knowledge about the clustering process, with different definitions leading to different clusterings. However, cluster definitions used in practice are often quite simple, for example clusters in $k$-means are defined in terms of $\ell_2$ distance to centroids, while Gaussians are a commonly used generative model for clusters in mixture models.

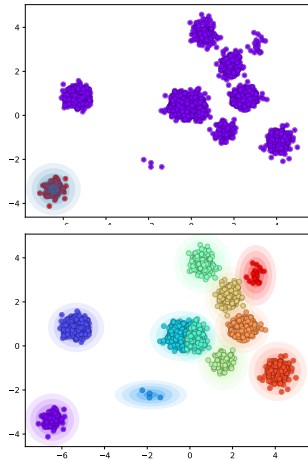

Figure 1: Our model identifies one cluster per iteration (top), allowing it to find any number of clusters (bottom).

Recently, advances in deep learning has facilitated the approximation of complex functions in a black-box fashion. One particular application of relevance to the problem of clustering in this paper is that of amortized inference (Gershman & Goodman, 2014; Stuhlmüller et al., 2013), where neural networks are trained to predict the states of latent variables given observations in a generative model or probabilistic programme. In the context of learning set-input neural networks (Zaheer et al., 2017), Lee et al. (2019) showed that it is possible to amortize the iterative clustering process for a Mixture of Gaussians (MoG), while Pakman et al. (2019) demonstrated that it is possible to train a neural network to sequentially assign data points to clusters. Both approaches can be interpreted as using neural networks for amortized inference of cluster assignments and parameters given a dataset. Note that once neural networks are used for amortized clustering, we can take advantage of their flexibility in working with more complex ways to define clusters. Further, the amortization networks can be trained using generated datasets where the ground truth clusterings are known. This can be interpreted as implicitly learning the definition of clusters underlying the training datasets, such that amortized inference (approximately) produces the appropriate clusterings. In a sense this shares a similar philosophy as Neural Processes (Garnelo et al., 2018b;a), which meta-learns from multiple datasets to learn a prior over functions.

In this paper, we build on these prior works and propose Deep Amortized Clustering (DAC). As in prior works, the amortization networks in DAC are trained using generated datasets where the ground truth clusterings are known. Like Lee et al. (2019), DAC uses a Set Transformer, but differs from Lee et al. (2019) in that it generates clusters sequentially, which enables to produce a varying number of clusters depending on the complexity of the dataset (Fig. 1). Our approach also extends Lee et al. (2019) from MOG to problems with more complex cluster definitions, which are arguably harder to hand specify and easier to meta-learn from data. Our work also differs from Pakman et al. (2019) in that our network processes data points in parallel while Pakman et al. (2019) processes them sequentially, which is arguably less scalable and limits applicability to smaller datasets.

This paper is organized as follows. We begin by describing in Section 2 the permutation-invariant set transformer modules that we use throughout the paper. In Section 3, we describe how we implement our core idea of identifying one cluster at a time, and describe our framework for clustering, the DAC There are several challenges in solving DAC on complex datasets, and we structured our paper roughly in order of difficulty. We apply DAC to clustering synthetic data (Section 5) and image data (Section 6); some settings required additional components, which we describe when needed.

## 2 A Primer on Set Transformer and Amortized Clustering

In this section, we briefly review the set-input neural network architectures to be used in the paper, and describe how Lee et al. (2019) used them to solve amortized clustering for MOG.

### 2.1 Set Transformer

The Set Transformer (ST) is a permutation-invariant set-input neural network that uses self-attetntion operations as building blocks. It utilizes multi-head attention (Vaswani et al., 2017) for both encoding elements of a set and decoding encoded features into outputs.

The fundamental building block of a ST is the Multihead Attention Block (MAB), which takes two sets $X = [x_1, \ldots, x_n]^\top$ and $Y = [y_1, \ldots, y_m]^\top$ and outputs a set of the same size as $X$. Throughout this article, we represent sets as matrices where each row corresponds to an element. An MAB is defined as

$$\text{MAB}(X, Y) = H + \text{rFF}(H) \text{ where } H = X + \text{rFF}(\text{MultiheadAtt}(X, Y)), \tag{1}$$

where $\text{rFF}(\cdot)$ is a feed-forward layer applied row-wise (i.e., for each element). $\text{MAB}(X, Y)$ computes the pairwise interactions between the elements in $X$ and $Y$ with sparse weights obtained from attention. A Self-Attention Block (SAB) is simply MAB applied to the set itself: $\text{SAB}(X) \triangleq \text{MAB}(X, X)$. We can model high-order interactions among the items in a set by stacking multiple SABs; we denote such a stack of $L$ SABs applied to set $X$ as $\text{SAB}_L(X)$.

To summarize a set into a fixed-length representation, ST uses an operation called Pooling by Multihead Attention (PMA). A PMA is defined as $\text{PMA}_k(X) = \text{MAB}(S, X)$ where $S = [s_1, \ldots, s_k]^\top$ are trainable parameters.

Note that the time-complexity of SAB is $O(n^2)$ because of pairwise computation. To reduce this, Lee et al. (2019) proposed to use Induced Self-Attention Block (ISAB) defined as

$$\text{ISAB}(X) = \text{MAB}(X, \text{MAB}(I, X)), \tag{2}$$

where $I = [i_1, \ldots, i_m]^\top$ are trainable *inducing points*. ISAB indirectly compares the elements of $X$ through the inducing points, reducing the time-complexity to $O(nm)$. Similarly to the SAB, we write $\text{ISAB}_L(X)$ to denote a stack of $L$ ISABs.

### 2.2 Amortized Clustering with Set Transformer

Lee et al. (2019) presented an example using ST for amortized inference for a MOG. A dataset $X$ is clustered by maximizing the likelihood of a $k$ component MOG, and a ST is used to output the parameters as:

$$H_X = \text{ISAB}_L(X), \; H_\theta = \text{PMA}_k(H_X), \; (\text{logit } \pi_j, \theta_j)_{j=1}^k = \text{rFF}(\text{SAB}_{L'}(H_\theta)), \tag{3}$$

where $\pi_j$ is the mixing coefficient and $\theta_j = (\mu_j, \sigma_j^2)$ are the mean and variance for the $j$th Gaussian component. The network is trained to maximize the expected log likelihood over *datasets*:

$$\mathbb{E}_{p(X)}\left[ \sum_{i=1}^{n_X} \log \sum_{j=1}^{k} \pi_j \log \mathcal{N}(x_i; \mu_j, \sigma_j^2) \right], \qquad (4)$$

where $n_X$ is the number of elements in $X$. Clustering is then achieved by picking the highest posterior probability component for each data point under the MoG with parameters output by the ST.

## 3 Deep Amortized Clustering

An apparent limitation of the model described in Section 2.2 is that it assumes a fixed number of clusters generated from Gaussian distributions. In this section, we describe our method to solve DAC in the more realistic scenario of having a variable number of clusters and arbitrarily complex cluster shapes.

### 3.1 Filtering: inferring one cluster at a time

The objective (4) is not applicable when the number of clusters is not fixed nor bounded. A remedy to this is to build a set-input neural network $f$ that identifies the clusters iteratively and make it to learn "when to stop", similar to Adaptive Computation Time (ACT) for RNNs (Graves, 2016).

One may think of several ways to implement this idea (we present an illustrative example that simply augments ACT to ST in Section 5.1). Here we propose to train $f$ to solve a simpler task - instead of clustering the entire dataset, focus on finding *one cluster at a time*. The task, what we call as *filtering*, is defined as a forward pass through $f$ that takes a set $X$ and outputs a parameter $\theta$ to describe a cluster along with a membership probability vector $\mathfrak{m} \in [0, 1]^{n_X}$ where $n_X$ is the number of elements in $X$. The meaning of the parameter $\theta$ depends on the specific problem. For example, $\theta$ for MoG is $(\mu, \sigma^2)$, the parameters of a Gaussian distribution. $\mathfrak{m}_i$ represents the probability of $x_i$ belonging to the cluster described by $\theta$. To filter out the datapoints that belong to the current cluster, we use $0.5$ as the threshold to discretize $\mathfrak{m}$ to a boolean mask vector. The resulting smaller dataset is then fed back into the neural network to produce the next cluster and so on.

**Minimum Loss Filtering**  Now we describe how to train the filtering network $f$. Assume $X$ has $k_x$ true clusters, and let $y \in [1, \ldots, k_x]^{n_X}$ be a cluster label vector corresponding to the true clustering of $X$. Then we define the loss function for one filtering iteration producing one $\theta$ and one $\mathfrak{m}$ as

$$\mathcal{L}(X, y, \mathfrak{m}, \theta) = \min_{j \in \{1, \ldots, k_X\}} \left( \frac{1}{n_X} \sum_{i=1}^{n_X} \mathrm{BCE}(\mathfrak{m}_i, \mathbb{1}_{\{y_i = j\}}) - \frac{1}{n_{X|j}} \sum_{i|y_i = j} \log p(x_i; \theta) \right), \quad (5)$$

where $n_{X|j} := \sum_{i=1}^{n_X} \mathbb{1}_{\{y_i = j\}}$, $\mathrm{BCE}(\cdot, \cdot)$ is the binary cross-entropy loss, and $p(x; \theta)$ is the density of $x$ under cluster parameterised by $\theta$. This loss encourages $\theta$ to describe the data distribution of a cluster, and $\mathfrak{m}$ to specify which datapoints belong to this particular cluster. The rationale to take minimum across the clusters is follows. One way to train $f$ to pick a cluster at each iteration is to impose an ordering on the clusters (e.g. in order of appearance in some arbitrary indexing of $X$, or in order of distance to origin), and to train $f$ to follow this order. However, this may introduce unnecessary inductive biases that deteriorates learning. Instead, we let $f$ find the *easiest* one to identify, thus promoting $f$ to learn its own search strategy. Note that there are $k_x!$ equally valid ways to label the clusters in $X$. This combinatorial explosion makes learning with standard supervised learning objectives for $y$ tricky, but our loss (5) is inherently free from this problem while being invariant to the labelling of clusters.

We use the following architecture for the filtering network $f$: Section 2:

$$\begin{array}{lll} \textbf{encode data:} & H_X = \mathrm{ISAB}_L(X), & \\ \textbf{decode cluster:} & H_\theta = \mathrm{PMA}_1(H_X), & \theta = \mathrm{rFF}(H_\theta), \\ \textbf{decode mask:} & H_\mathfrak{m} = \mathrm{ISAB}_{L'}(\mathrm{MAB}(H_X, H_\theta)), & \mathfrak{m} = \mathrm{sigmoid}(\mathrm{rFF}(H_\mathfrak{m})). \end{array} \qquad (6)$$

The network first encodes $X$ into $H_X$ and extracts cluster parameters $\theta$. Then $\theta$ together with encoded data $H_X$ are further processed to produce the membership probabilities $\mathfrak{m}$. We call the filtering network with architecture (6) and trained with objective (5) Minimum Loss Filtering (MLF).

**Anchored Filtering** An alternative strategy that we found beneficial for harder datasets is to use *anchor points*. Given a dataset $X$ and labels $y$ constructed from the true clustering, we sample an anchor point with index $a \in \{1, \ldots, n_x\}$ uniformly from $X$. We parameterize a set-input network $f$ to take both $X$ and $a$ is input, and to output *the cluster that contains the anchor point $x_a$*. The corresponding loss function is,

$$\mathcal{L}(x, y, a, \mathfrak{m}, \theta) = \frac{1}{n_X} \sum_{i=1}^{n_X} \text{BCE}(\mathfrak{m}_i, \mathbb{1}_{\{y_i=j_a\}}) - \frac{1}{n_{X|j_a}} \sum_{i|y_i=j_a} \log p(x_i; \theta), \quad (7)$$

where $j_a$ denotes the the true cluster index containing $a$. The architecture to be trained with this loss can be implemented as

$$
\begin{aligned}
\textbf{encode data:} \quad & H_X = \text{ISAB}_L(X), & H_{X|a} = \text{MAB}(H_X, h_a), \\
\textbf{decode cluster:} \quad & H_\theta = \text{PMA}_1(H_{X|a}), & \theta = \text{rFF}(H_\theta), \\
\textbf{decode mask:} \quad & H_\mathfrak{m} = \text{ISAB}_{L'}(\text{MAB}(H_{X|a}, H_\theta)), & \mathfrak{m} = \text{sigmoid}(\text{rFF}(H_\mathfrak{m})), \quad (8)
\end{aligned}
$$

where $h_a$ is the row vector of $H_X$ corresponding to the index $a$. We train (8) by randomly sampling $a$ for each step, and thus promoting $f$ to find clusters by comparing each data point to the random anchor point. Note that the loss is also free from the label order ambiguity given anchor points. We call this filtering strategy Anchored Filtering (AF).

## 3.2 Beyond Simple Parametric Families

When each cluster cannot be well described by a Gaussian or other simple parametric distributions, we have several choices to learn them. The first is to estimate the densities along with the filtering using neural density estimators such as Masked Autoregressive Flow (MAF) (Papamakarios et al., 2017). Another option is to lower-bound $\log p(x; \theta)$ by introducing variational distributions, for example using Variational Autoencoder (VAE) (Kingma & Welling, 2014). See Section 5.2 and Section 6.1 for examples. If the density estimation is not necessary, we can choose not to learn $\log p(x; \theta)$. In other words, instead of (5) and (7), we train

$$\mathcal{L}(X, y, \mathfrak{m}, \theta) = \min_j \sum_{i=1}^{n_X} \text{BCE}(\mathfrak{m}_i, \mathbb{1}_{\{y_i=j\}}), \quad \mathcal{L}(X, y, a, \mathfrak{m}, \theta) = \sum_{i=1}^{n_X} \text{BCE}(\mathfrak{m}_i, \mathbb{1}_{\{y_i=j_a\}}), \quad (9)$$

for MLF and AF, respectively. The corresponding architectures are (6) and (8) with parameter estimation branches removed. The DAC trained in this way implicitly learns how to define a cluster from the given training datasets and cluster labels. See Section 6.2 and Section 6.3 where we applied this to cluster image datasets.

## 3.3 Deep Amortized Clustering

Recall that each step of filtering yields one cluster. To solve DAC, we iterate this procedure until all clusters are found. After each filtering step, we remove from the dataset the points that were assigned to the cluster, and perform filtering again. This recursive procedure is repeated until all datapoints have been assigned to a cluster[1]. We call the resulting amortized clustering algorithm as DAC[2]. DAC learns both data generating distributions and cluster assignment distributions from meta-training datasets without explicit hand-engineering.

## 4 Related Works

**Deep clustering methods** There is a growing interest in developing clustering methods using deep networks for complex data (Yang et al., 2016a;b; Xie et al., 2015; Li et al., 2017; Ji et al., 2018). See Aljalbout et al. (2018) for a comprehensive survey on this line of work. The main

---

[1] In practice, we input the entire dataset along with $\mathfrak{m}$ and assign zero attention weight to datapoints with $\mathfrak{m}_i = 1$. This is equivalent to the described scheme, but has the added benefit of being easy to parallelize across multiple datasets.

[2] Note that DAC with MLF is not stochastic once we discretize the membership probability.

focus of these methods is to learn a representation of input data amenable to clustering via deep neural networks. Learning representations and assigning data points to the clusters are usually trained alternatively. However, like the traditional clustering algorithms, these methods aim to cluster particular datasets. Since such methods typically learn a data representation using deep neural networks, the representation is prone to overfitting when applied to small datasets.

**Learning to Cluster** Learning to cluster refers to the task of learning a clustering algorithm from data. Such methods are trained in a set of source datasets and tested on unseen target datasets. Constrained Clustering Networks (Hsu et al., 2017; 2019) follow a two-step process for learning to cluster: they first learn a similarity metric that predicts whether a given pair of datapoints belong to the same class, and then optimize a neural network to predict assignments that agree with the similarity metric. Centroid Networks (Huang et al., 2019) learn an embedding which is clustered with the Sinkhorn K-means algorithm. While these methods combine deep networks with an iterative clustering algorithm, our framework is much more efficient as it directly identifies each cluster after one forward pass. Our experiments in Section 6 that our model is orders of magnitude faster than previous works in learning to cluster.

**Amortized clustering methods** To the best of our knowledge, the only works that consider a similar task to ours is Lee et al. (2019) and Pakman et al. (2019). We refer the reader back to Section 2 for an outline of the amortized clustering framework presented in Lee et al. (2019). Pakman et al. (2019) presented an amortized clustering method called Neural Clustering Process (NCP). Given a dataset, NCP sequentially computes the conditional probability of assigning the current data point to one of already constructed clusters or a new one, similar in spirit to the popular Gibbs sampling algorithm for Dirichlet process mixture models (Neal, 2000), but without positing particular priors on partitions, but rather letting the network learn from data. However, the sequential sampling procedure makes the algorithm not parallizable using modern GPUs, limiting its scalability. Furthermore, since the clustering results vary a lot w.r.t. the sequential processing order, the algorithm needs a sufficient number of random samples to get stable clustering results. We compared our method to NCP on small-scale MoG experiments in Appendix C, and our results support our claim.

## 5 EXPERIMENTS ON SYNTHETIC DATASETS

### 5.1 2D MIXTURE OF GAUSSIANS

We first demonstrate DAC with MLF on 2D MoG datasets with arbitrary number of clusters. We considered two baselines that can handle variable number of clusters: truncated Variational Bayesian Dirichlet Process Mixture Model (VBDPM) (Blei & Jordan, 2006) and the ST architecture (Lee et al., 2019) with ACT-style decoder so that it can produce arbitrary number of clusters. We describe the latter method, ACT-ST, in detail in Appendix A. See Appendix B for detailed experimental setup including data generation process and training scheme. We trained DAC and ACT-ST using random datasets with a random number of data points $n \leq n_{max}$ and clusters $k \leq k_{max}$, where we set $(n_{max}, k_{max}) = (1000, 4)$ during training. We tested the resulting model on two scenarios. The first one is to test one same configurations; testing on 1,000 random datasets with $(n_{max}, k_{max}) = (1000, 4)$. The second one is test on 1,000 random datasets with $(n_{max}, k_{max}) = (3000, 12)$ to see whether the amortized clustering methods can generalize to an unseen number of clusters. For both scenarios, we ran VBDPM on *each test dataset* until convergence from scratch. We used the mean of the variational distributions as the point-estimates of the parameters to compute log-likelihoods. Table 1 summarizes the results. DAC works well for both cases, even beating the oracle log-likelihood computed from the true parameters, while ACT-ST fails to work in the more challenging $(n_{max}, k_{max}) = (3000, 12)$ case (Fig. 2). VBDPM works well for both, but it takes considerable time to converge whereas DAC requires no such optimization to cluster test datasets.

### 5.2 2D MIXTURE OF WARPED GAUSSIANS

When the parametric form of cluster distribution is not known, we cannot directly compute the $\log p(x_i; \theta)$ term in (5). In this case, we propose to estimate the density $p(x; \theta)$ via neural density estimators along with the DAC learning framework. We construct each cluster by first sampling points from a 2D unit Gaussian distribution, and then applying a random nonlinear transformation on

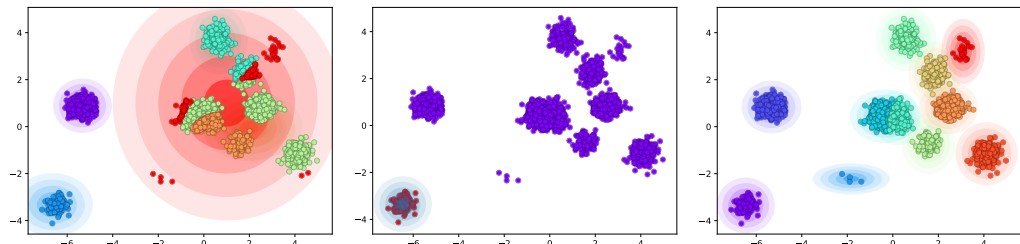

Figure 2: (Left) Clustering by ACT-ST, (middle) one step of filtering, (right) clustering by DAC.

Table 1: Results on synthetic 2D MoG. We report log-likelihood (LL), clustering accuracies (adjusted Rand index (ARI) (Hubert & Arabie, 1985) and normalized mutual information (NMI), the mean absolute error between true $k$ and estimated $k$, and processing time per dataset. The numbers below $(n_{max}, k_{max})$ are oracle LL values computed by the true parameters. We report the average on 5 runs.

| $(n_{max}, k_{max})$ | Algorithm | LL | ARI | NMI | $k$-MAE | Time [sec] |
|---|---|---|---|---|---|---|
| (1000,4) -0.693 | VBMOG | -0.719 ± 0.002 | 0.971 ± 0.001 | 0.977 ± 0.001 | 0.079 ± 0.003 | 0.037 ± 0.001 |
| | ACT-ST | -0.721 ± 0.008 | 0.974 ± 0.002 | 0.974 ± 0.001 | **0.044** ± 0.003 | **0.006** ± 0.000 |
| | DAC | **-0.692** ± 0.002 | **0.983** ± 0.001 | **0.978** ± 0.001 | 0.120 ± 0.009 | 0.008 ± 0.000 |
| (3000,12) -1.527 | VBMOG | -1.561 ± 0.001 | 0.962 ± 0.000 | 0.970 ± 0.000 | 0.435 ± 0.010 | 0.400 ± 0.006 |
| | ACT-ST | -5.278 ± 0.573 | 0.781 ± 0.008 | 0.855 ± 0.004 | 1.993 ± 0.082 | 0.024 ± 0.001 |
| | DAC | **-1.544** ± 0.006 | **0.971** ± 0.000 | **0.974** ± 0.000 | **0.279** ± 0.012 | **0.021** ± 0.001 |

each point. We use MAF (Papamakarios et al., 2017) to model $p(x; \theta)$ where $\theta$ is a context vector that summarizes the information about a cluster. We trained the resulting DAC with MLF using random datasets, and compared to spectral clustering (Shi & Malik, 2000). As shown in Fig. 3, DAC finds and estimates the densities of these nonlinear clusters. See Appendix D for more details and results.

## 6 EXPERIMENTS ON REAL DATASETS

### 6.1 CLUSTERING EMNIST WITH MIXTURE OF NEURAL STATISTICIANS

We may approximate the likelihood $p(x; \theta)$ via a VAE (Kingma & Welling, 2014) when the data distribution is too high-dimensional or complex. Instead of directly maximizing the log-likelihood, we maximize a lower-bound on the likelihood, $\log p(x; \theta) \geq \mathbb{E}_{q(z|x;\theta)}[\log p(x, z; \theta) - q(z|x; \theta)]$, where $\theta$ encodes the context of a cluster and $z$ is a latent variable that describes $x$ based on $\theta$. Neural Statistician (NS) (Edwards & Storkey, 2016) proposed the idea of approximating $p(x; \theta)$ using a context $\theta$ produced by a set network; we thus call this model a mixture of NSs. We found that the DAC implemented in this way could cluster well, generalizes to an unseen number of clusters, and generate images conditioned on the cluster context $\theta$. See Appendix E for detailed results.

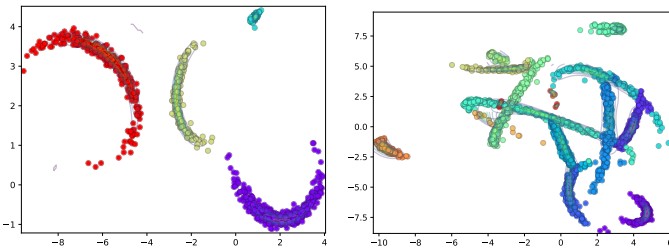

Figure 3: Clustering warped Gaussian data with DAC. The model was trained with $k \in \{1, \ldots, 4\}$ clusters (left) but generalizes to datasets with 12 clusters (right).

Table 2: Results for 1,000 test datasets sampled from Embedded ImageNet.

|  | ARI | NMI | $k$-MAE | Time (sec) |
|---|---|---|---|---|
| $k$-means | $0.370 \pm 0.001$ | $0.514 \pm 0.001$ | - | $0.188 \pm 0.003$ |
| Spectral (Shi & Malik, 2000) | $0.432 \pm 0.000$ | $0.568 \pm 0.000$ | - | $0.087 \pm 0.002$ |
| DEC (Xie et al., 2015) | 0.195 | 0.326 | - | 46.098 |
| KCL (Hsu et al., 2017) | 0.201 | 0.361 | - | 13.401 |
| MCL (Hsu et al., 2019) | 0.157 | 0.350 | - | 14.646 |
| DAC$_{\text{MLF}}$ | $0.400 \pm 0.012$ | $0.527 \pm 0.013$ | $2.103 \pm 0.160$ | $0.012 \pm 0.001$ |
| DAC$_{\text{AF}}$ | $\mathbf{0.451} \pm 0.014$ | $\mathbf{0.579} \pm 0.013$ | $1.805 \pm 0.031$ | $\mathbf{0.017} \pm 0.001$ |

Table 3: Unsupervised cross-task transfer learning on Omniglot. Normalized mutual information (higher is better) is averaged across 20 alphabets (datasets), each of which have between 20 and 47 letters (classes). All values beside DAC were reported in Hsu et al. (2019). "$k$ given" means that the true number of clusters was given to the model.

| Method | NMI($k$ given) | NMI |
|---|---|---|
| K-means (MacQueen et al., 1967) | 0.353 | 0.464 |
| LPNMF (Cai et al., 2009) | 0.372 | 0.498 |
| LSC (Chen & Cai, 2011) | 0.376 | 0.500 |
| CSP (Wang et al., 2014) | 0.812 | 0.812 |
| MPCK-means (Bilenko et al., 2004) | 0.871 | 0.816 |
| KCL (Hsu et al., 2017) | 0.889 | 0.874 |
| MCL (Hsu et al., 2019) | **0.897** | **0.893** |
| DAC$_{\text{AF}}$ (ours) | n/a | **0.829** |

## 6.2 CLUSTERING EMBEDDED IMAGENET

We applied DAC to cluster the collection of miniImageNet (Vinyals et al., 2016) and tieredImageNet (Ren et al., 2018). We gathered pretrained 640 dimensional features of the images released by Rusu et al. (2018)[3]. We used the training and validation features as training set and test features as test set. The resulting training set contains 620,000 samples from 495 classes, and the test set contains 218,000 samples from 176 classes, with no overlap between training and test classes.

We trained DAC without density estimations (9), using both MLF and AF. We sampled randomly clustered datasets from the training set having $(n_{\max}, k_{\max}) = (100, 4)$. We then generated 1,000 randomly clustered datasets from the test set with $(n_{\max}, k_{\max}) = (300, 12)$. We compared DAC to basic clustering algorithms ($k$-means, spectral clustering), Deep Embedding Clustering (DEC) (Xie et al., 2015), and transfer learning methods (KCL (Hsu et al., 2017) and MCL (Hsu et al., 2019)). $k$-means, spectral, and DEC were trained for each test dataset from scratch. For KCL and MCL, we first trained a similarity prediction network using the training set, and used it to cluster each test dataset. For DEC, KCL and MCL, we used fully-connected layers with 256 hidden units and 3 layers for both similarity prediction and clustering network. Note that the size of the test datasets are small ($n_{\max} = 300 < 640$), so one can easily predict that DEC, KCL, and MCL would overfit. The algorithms other than ours was given the true number of clusters. The results are summarized in Table 2. It turns out that the pretrained features are good enough for the basic clustering algorithms to show decent performance. The deep learning based methods failed to learn useful representations due to the small dataset sizes. Ours showed the best clustering accuracies while also consuming the shortest computation time.

## 6.3 CLUSTERING OMNIGLOT IMAGES

We apply our filtering architecture to the unsupervised cross-task transfer learning benchmark of Hsu et al. (2017; 2019). This benchmark uses the Omniglot dataset (Lake et al., 2015) to measure how well a clustering method can generalize to unseen classes. The Omniglot dataset consists of

---

[3]https://github.com/deepmind/leo

Table 4: Mean absolute error of cluster number ($k$) estimate and processing time per dataset on the Omniglot benchmark.

| | $k$-MAE | | | Time [sec] | |
|---|---|---|---|---|---|
| KCL | MCL | $DAC_{AF}$ | KCL | MCL | $DAC_{AF}$ |
| $6.4 \pm 6.4$ | $5.1 \pm 4.6$ | $\mathbf{4.6} \pm \mathbf{2.7}$ | $129.3 \pm 18.9$ | $124.5 \pm 14.4$ | $\mathbf{4.3} \pm \mathbf{0.6}$ |

handwritten characters from 50 different alphabets. Each alphabet consists of several characters, and each character has 20 images drawn by different people. Our problem setup consists of training a clustering model using images from the 30 background alphabets, and using each of the 20 alphabets as a seperate dataset to test on. We use the same VGG network backbone as in Hsu et al. (2019) and follow their experimental setup.

We show the normalized mutual information (NMI) of DAC along with other methods in Table 3. While previous methods were also evaluated on the easier setting where the true number of clusters is given to the network, this setting is not applicable to DAC. We see that DAC is competitive with the state-of-the-art on this challenging task despite requiring orders of magnitude less computation time. While the metrics for DAC were computed after at most 100 forward passes, previous methods all require some sort of iterative optimization. For example, KCL and MCL (Hsu et al., 2017; 2019) required more than 100 *epochs* of training *for each alphabet* to arrive at the cluster assignments in Table 3. This difference in computation requirements is more clearly demonstrated in Table 4: DAC requires an average of less than 5 seconds per alphabet whereas KCL and MCL required more than 100. Table 4 additionally shows that in addition to being extremely time-efficient, DAC was more accurate in estimating $k$. This demonstrates the efficacy of our overall structure of identifying one cluster at a time.

## 7 DISCUSSION AND FUTURE WORK

We have proposed DAC, an approach to amortized clustering using set-input neural networks. DAC learns to cluster from data, without the need for specifying the number of clusters or the data generating distribution. It clusters datasets efficiently, using a few forward passes of the dataset through the network.

There are a number of interesting directions for future research. The clustering results produced by DAC is almost deterministic because we discretise the membership probabilities in the filtering process. It would be interesting to take into account uncertainties in cluster assignments. In the imagenet experiment, we found that we needed a sufficient number of training classes to make DAC generalize to unseen test classes. Training DAC to generalize to unseen image classes with smaller numbers of training classes seem to be a challenging problem. Finally, learning DAC along with state-of-the-art density estimation techniques for images in each cluster is also a promising research direction.

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

## A    DETAILED DESCRIPTION OF ACT-ST

We first define an adaptive-computation-time version of PMA,

$$\text{aPMA}(X, k) = \text{MAB}([s_1, \ldots, s_k]^\top, X), \quad s_j = \text{PMA}_1([s_1, \ldots, s_{j-1}]^\top) \text{ for } j = 2, \ldots, k, \tag{10}$$

which enables an RNN-like iterative computation by sequentially extending the parameters for PMA. The clustering network to output variable number of parameters is then defined as

$$H_X = \text{ISAB}_L(X), \quad H_\theta^{(k)} = \text{SAB}_{L'}(\text{aPMA}(H, k)),$$

$$v_k = \text{sigmoid}(\text{mean}(\text{rFF}(H_\theta^{(k)}[:,1]))), \quad s_k = 1 - \prod_{j \leq k} v_k,$$

$$(\text{logit } \pi_j^{(k)}, \theta_j^{(k)})_{j=1}^k = \text{rFF}(H_\theta^{(k)}[:,2:]), \tag{11}$$

where $[:,1]$ and $[:,2:]$ are numpy-like notation indexing the columns. $s_k$ is a "stop" variable where $s_k > 0.5$ means the iteration stops at $k$th step and continues otherwise.

During training, we utilize the true number of clusters as supervision for training $c_k$, yielding the overall loss function

$$\mathbb{E}_{p(X, k_{\text{true}})} \left[ -\sum_{i=1}^{n_X} \log \sum_{j=1}^{k_{\text{true}}} \pi_j^{(k_{\text{true}})} p(x_i; \theta_j^{(k_{\text{true}})}) + \sum_{k=1}^{k_{\text{max}}} \text{BCE}(c_k, \mathbb{1}_{\{k < k_{\text{true}}\}}) \right]. \tag{12}$$

where $k_{\text{true}}$ is the true number of clusters, $k_{\text{max}} \geq k_{\text{true}}$ is maximum number of steps to run.

## B    DETAILS OF MOG EXPERIMENTS

We generated dataset by the following process.

$$n \sim \text{Unif}(0.3n_{\text{max}}, n_{\text{max}}), \quad k - 1 \sim \text{Binomial}(k_{\text{max}} - 1, 0.5),$$

$$\pi \sim \text{Dir}(\alpha \overbrace{[1, \ldots, 1]}^{k}), \quad (y_i)_{i=1}^n \overset{\text{i.i.d.}}{\sim} \text{Cat}(\pi)$$

$$(\mu_j)_{j=1}^k \overset{\text{i.i.d.}}{\sim} \text{Normal}([0,0]^\top, 9I), \quad (\sigma_j)_{j=1}^k \overset{\text{i.i.d.}}{\sim} \log \text{Normal}(\log(0.25)[1,1]^\top, 0.01I),$$

$$x_i \sim \text{Normal}(\mu_{y_i}, \text{diag}(\sigma_{y_i}^2)) \text{ for } i = 1, \ldots, n. \tag{13}$$

Both ACT-ST and filtering networks were trained with $n_{\text{max}} = 1,000$ and $k_{\text{max}} = 4$. For each step of training, we sampled a batch of 100 datasets (sharing the same $n \sim \text{Unif}(0.3n_{\text{max}}, n_{\text{max}})$ to comprise a tensor of shape $100 \times n \times 2$), and computed the stochastic gradient to update parameters. We trained the networks for 20,000 steps using ADAM optimizer (Kingma & Ba, 2015) with initial learning rate $5 \times 10^{-4}$. The results in Table 1 are obtained by testing the trained models on randomly generated 1,000 datasets with the same generative process.

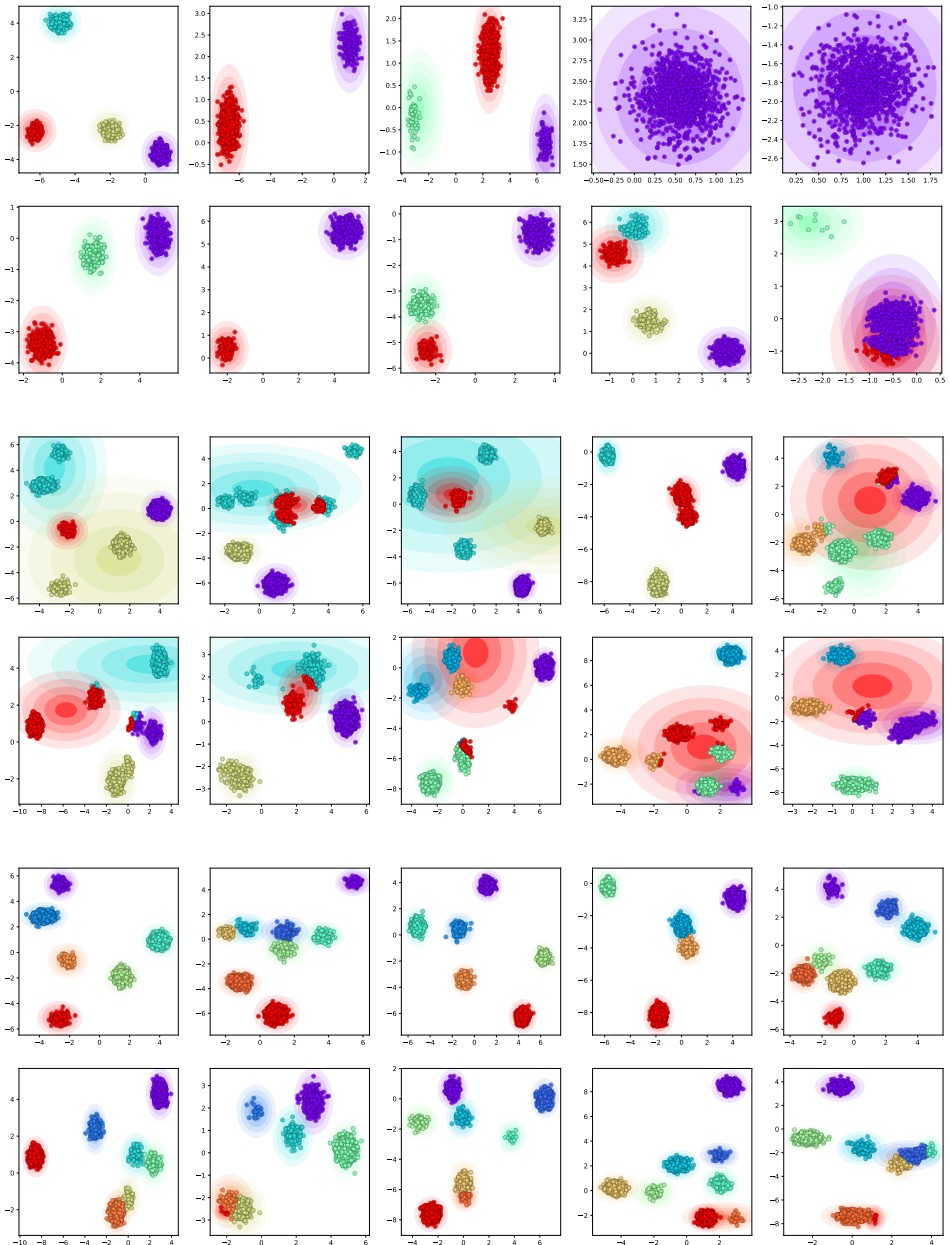

Figure 4: More clustering results. Top two rows shows ACT-ST applied to the datasets having $k \in \{1, \ldots, 4\}$. The middle two rows show the same ACT-ST model applied to datasets having $k > 4$. The bottom two rows show iterative filtering applied to the same datasets.

Table 5: Comparison of iterative filtering and NCP on 100 random datasets with $(n_{\max}, k_{\max}) = (3000, 12)$. The oracle log-likelihood is $-1.5309$, and DAC recorded $-1.5640$. $S$ is the number of samples per dataset used for NCP.

|  | ARI | $k$-MAE | Time [sec] |
|---|---|---|---|
| DAC | 0.9616 | 0.2800 | 0.0208 |
| NCP ($S = 1$) | 0.7947 | 1.6333 | 4.1435 |
| NCP ($S = 10$) | 0.8955 | 0.8000 | 5.8920 |
| NCP ($S = 50$) | 0.9098 | 0.6444 | 6.7936 |

## C  COMPARISON TO NEURAL CLUSTERING PROCESS

We compare DAC to NCP. Due to the sequential nature, the training procedure of NCP does not scale to the other experiments we conducted. Instead, we trained NCP for MoG data described in Appendix B with smaller scale having $n_{\max} = 100$ and $k_{\max} = 4$. We used the code released by the authors[4] with default hyperparameters. We measured the clustering performance for 100 random datasets generated with $(n_{\max}, k_{\max}) = (3000, 12)$ (Table 5). Both method generalized well w.r.t. the number of data points $n$, but filtering did much better in generalizing for the number of clusters $k$. The performance of NCP for clustering depends heavily on the processing order, so we conducted multiple runs with different random orders and picked the best one w.r.t. the clustering probability computed from NCP. Ours outperformed NCP with $S = 50$ samples per dataset with at least two orders of magnitude faster processing time.

## D  DETAILS OF MIXTURE OF MAFs AND WARPED GAUSSIAN EXPERIMENTS

We model the cluster density as MAF.

$$
\begin{aligned}
\log p(x; \theta) &= \log p(x_1) + \sum_{i=2}^{d} \log p(x_i | x_{1:i-1}; \theta) \\
&= \log \mathrm{Normal}(x_1 | 0, 1) + \sum_{i=2}^{d} \log \mathrm{Normal}(x_i | \mu(x_{1:i-1}, \theta), \sigma^2(x_{1:i-1}, \theta)) \\
&= \log \mathrm{Normal}(u | 0_d, I_d) - \sum_{i=2}^{d} \log \sigma(x_{1:i-1}, \theta),
\end{aligned}
\tag{14}
$$

where

$$
u_1 = x_1, \quad u_i = \frac{x_i - \mu_i(x_{1:i-1}, \theta)}{\sigma(x_{1:i-1}, \theta)}.
\tag{15}
$$

We can efficiently implement this with MADE (Germain et al., 2015).

---

[4]https://github.com/aripakman/neural_clustering_process

Table 6: Results on warped Gaussian datasets.

| $(n_{\max}, k_{\max})$ | Algorithm | LL | ARI | NMI | $k$-MAE | Time [sec] |
|---|---|---|---|---|---|---|
| (1000,4) | Spectral | - | $0.845 \pm 0.003$ | $0.889 \pm 0.002$ | - | $0.103 \pm 0.000$ |
|  | DAC | $-1.275 \pm 0.015$ | $0.974 \pm 0.001$ | $0.970 \pm 0.001$ | $0.320 \pm 0.035$ | $0.011 \pm 0.000$ |
| (3000,12) | Spectral | - | $0.592 \pm 0.001$ | $0.766 \pm 0.001$ | - | $0.572 \pm 0.003$ |
|  | DAC | $-2.436 \pm 0.029$ | $0.923 \pm 0.002$ | $0.936 \pm 0.001$ | $1.345 \pm 0.099$ | $0.037 \pm 0.001$ |

We generated the warped Gaussian datasets by the following generative process.

$$n \sim \text{Unif}(0.3n_{\max}, n_{\max}), \quad k - 1 \sim \text{Binomial}(k_{\max} - 1, 0.5),$$

$$\pi \sim \text{Dir}(\alpha \overbrace{[1, \ldots, 1]}^{k}), \quad (y_i)_{i=1}^{n} \overset{\text{i.i.d.}}{\sim} \text{Cat}(\pi)$$

$$\tilde{r} \sim \text{MoG}_1(y), \quad r = 0.8\pi\tilde{r},$$

$$(a_j)_{j=1}^{k} \overset{\text{i.i.d.}}{\sim} \text{Normal}(0, \sqrt{2}), \quad (b_j)_{j=1}^{k} \overset{\text{i.i.d.}}{\sim} \text{Normal}(0, \sqrt{2})$$

$$s_i = a_{y_i} \cos r_i + 0.1 \frac{b_{y_i} \cos r_i}{\sqrt{a_{y_i}^2 + b_{y_i}^2}}, \quad t_i = b_{y_i} \sin r_i + 0.1 \frac{a_{y_i} \sin r_i}{\sqrt{a_{y_i}^2 + b_{y_i}^2}},$$

$$(\varrho_j)_{j=1}^{k} \overset{\text{i.i.d.}}{\sim} \text{Unif}(0, 2\pi), \quad R_i = \begin{bmatrix} \cos \varrho_{y_i} & -\sin \varrho_{y_i} \\ \sin \varrho_{y_i} & \cos \varrho_{y_i} \end{bmatrix},$$

$$(\lambda_j)_{j=1}^{k} \overset{\text{i.i.d.}}{\sim} \text{Normal}(\min(k, 4.0)[1, 1]^{\top}, I), \quad x_i = R_i[s_i, t_i]^{\top} + \lambda_{y_i}, \tag{16}$$

where $\text{MoG}_1(y)$ denotes the sampling from 1d Mixture of Gaussians with the same parameter distributions as (13).

The filtering network is constructed as (6) where $\theta$ is a 128 dimensional vector to be fed into MAF as a context vector for a cluster. We implemented $\log p(x; \theta)$ as a 4 blocks of MAF with MADE (Germain et al., 2015). The filtering network was trained using random datasets with $n_{\max} = 1,000$ and $k_{\max} = 4$, and trained for 20,000 steps with ADAM optimizer. We set initial learning rate as $5 \cdot 10^{-4}$. Table 6 compares the resulting DAC to spectral clustering (Shi & Malik, 2000). Spectral clustering was ran for each dataset from scratch with true number of clusters given. To give a better idea how good is the estimated log-likelihood values, we trained MAF with same structure (4 blocks of MADE) but without mixture component and cluster context vectors for 100 random test datasets. We trained for 20,000 steps for each dataset using ADAM optimizer with learning rate $5 \cdot 10^{-4}$. The log-likelihood values estimated with filtering on the same datasets outperforms the one obtained by exhaustively training MAF for each dataset, and got -2.570 for MAF and -2.408 for DAC. This shows that the amortized density estimation works really well.

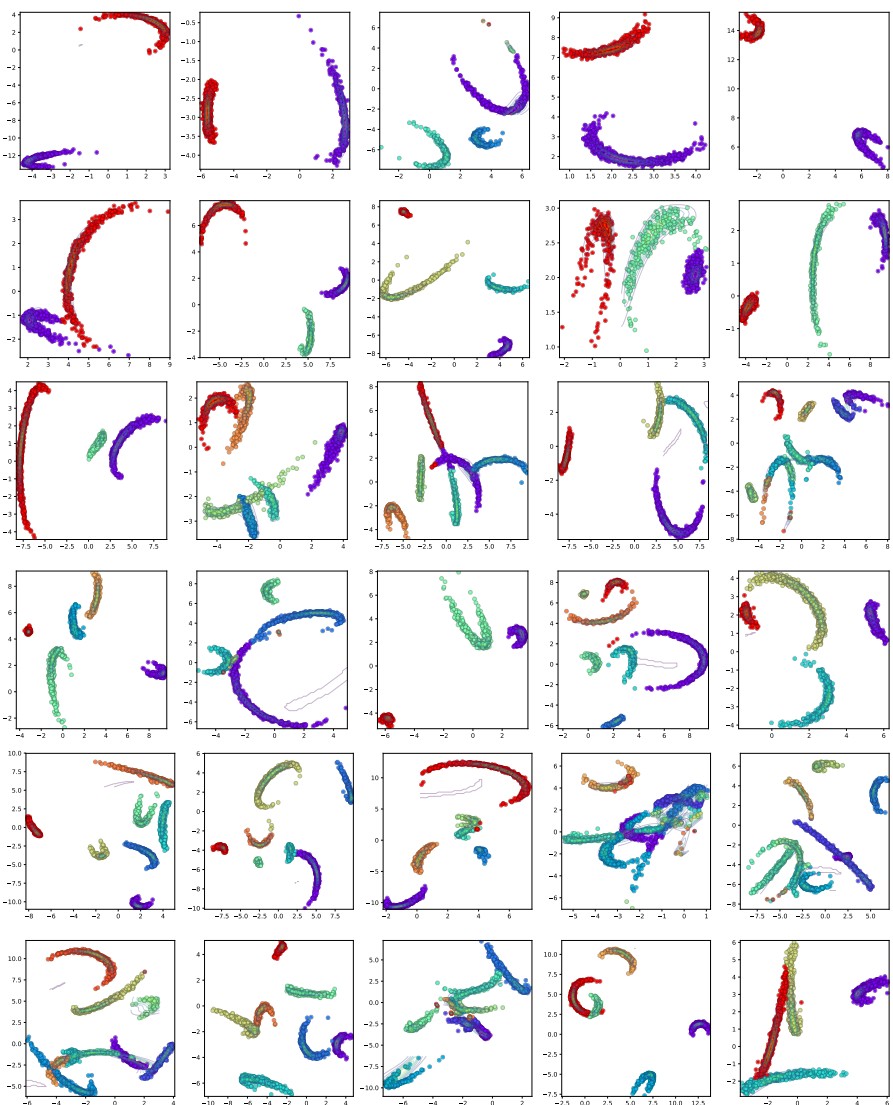

Figure 5: More clustering results for warped Gaussian datasets.

# E  DETAILS OF MIXTURE OF NEURAL STATISTICIANS AND EMNIST EXPERIMENTS

## E.1  MIXTURE OF NEURAL STATISTICIANS

Let $X = [x_1, \ldots, x_n]^\top$ be an input set. We construct a filtering network as follows.

$$
\begin{aligned}
&h_{\text{enc},i} = \text{Encoder}(x_i) \text{ for } i = 1, \ldots, n, \quad H_X = \text{ISAB}_L([h_{\text{enc},1}, \ldots, h_{\text{enc},n}]^\top), \\
&H_\theta = \text{PMA}_1(H_X), \quad \theta = \text{rFF}(H_\theta), \quad z_i \sim q(z_i|h_{\text{enc},i}, \theta) \text{ for } i = 1, \ldots, n \\
&h_{\text{dec},i} = \text{Decoder}(z_i, \theta) \text{ for } i = 1, \ldots, n, \quad \tilde{x}_i \sim p(x_i|h_{\text{dec},i}) \text{ for } i = 1, \ldots, n \\
&H_{\mathfrak{m}} = \text{ISAB}_L(\text{MAB}(H_X, H_\theta)), \quad \mathfrak{m} = \text{sigmoid}(\text{rFF}(H_{\mathfrak{m}})).
\end{aligned}
\tag{17}
$$

The log-likelihood for a particular cluster $\log p(\{x_i|y_i = j\}; \theta)$ is then approximate by the variational lower-bound,

$$
\begin{aligned}
\log p(\{x_i|y_i = j\}; \theta) &\geq \sum_{i|y_i=j} \int q(z_i|h_{\text{enc},i}, \theta) \log \frac{p(x_i|h_{\text{dec},i})p(z_i)}{q(z_i|h_{\text{enc},i}, \theta)} \mathrm{d}z_i \\
&= \sum_{i|y_i=j} \Big( \mathbb{E}_{q(z_i|h_{\text{enc},i},\theta))}[\log p(x_i|h_{\text{dec},i})] - \text{KL}[q(z_i|h_{\text{enc},i}, \theta)\|p(z_i)] \Big).
\end{aligned}
\tag{18}
$$

Once the clustering is done, the likelihood of the whole dataset $X$ can be lower-bounded as

$$
\log p(X; \theta_1, \ldots, \theta_k) \geq \sum_{j=1}^{k} \pi_j \log p(\{x_i|y_i = j\}; \theta_j).
\tag{19}
$$

The likelihood lower-bounding for each cluster corresponds to the NS except for that the context vector is constructed with ISAB and PMA, so we call this model a mixture of NSs.

## E.2  EXPERIMENTS ON EMNIST DATA

We trained the model described in (17) to EMNIST (Cohen et al., 2017). We picked "balanced" split, with 47 class / 112,800 training images / 18,800 test images. At each training step, we generated 10 randomly clustered dataset with $(n_{\max}, k_{\max}) = (1000, 4)$ and trained the network by the loss function (5) with the log-likelihood part replaced by (18). For $\text{Encoder}(x_i)$ and $\text{Decoder}(z_i, \theta)$, we used three layers of multilayer perceptrons. For the variational distribution $q(z_i|h_{\text{enc},i}, \theta)$, we used Gaussian distribution with parameters constructed by a fully-connected layer taking $[h_{\text{enc},i}, \theta]$ as inputs. Following Chen et al. (2017), We used an autoregressive prior distribution constructed by MAF for $p(z)$. For the likelihood distribution, $p(x|h_{\text{dec},i})$, we used Bernoulli distribution. Each training image was stochastically binarized.

Table 7 shows that iterative filtering can decently cluster EMNIST. Fig. 6 shows that given a set of 100 images with 4 clusters, the filtering can correctly identify a cluster and learns the generative model to describe it. Fig. 7 shows the clustering results using iterative filtering for 100 images.

Table 7: The results on EMNIST.

| $(n_{\max}, k_{\max})$ | LL/pixel | ARI | $k$-MAE |
|---|---|---|---|
| (1000,4) | $-0.193 \pm 0.002$ | $0.887 \pm 0.005$ | $0.607 \pm 0.093$ |
| (3000,12) | $-0.199 \pm 0.003$ | $0.728 \pm 0.019$ | $1.668 \pm 0.147$ |

# F  MORE RESULTS FOR OMNIGLOT EXPERIMENTS

We present more detailed comparion of KCL, MCL and ours for each alphabet.

Table 8: Absolute error of cluster number ($k$) estimate and time per dataset on the Omniglot benchmark. We show averages on the bottom row. Lower is better for both metrics.

| Alphabet | $k$ | Absolute Error of $k$ | | | Time (s) | | |
|---|---|---|---|---|---|---|---|
| | | KCL | MCL | DAC$_{AF}$ | KCL | MCL | DAC$_{AF}$ |
| Angelic | 20 | 6 | 2 | 2 | 110.6 | 102.5 | 3.3 |
| Atemayar Q. | 26 | 8 | 0 | 5 | 116.2 | 115.0 | 3.5 |
| Atlantean | 26 | 15 | 1 | 10 | 115.5 | 112.5 | 4.1 |
| Aurek_Besh | 26 | 2 | 4 | 1 | 190.2 | 113.2 | 2.9 |
| Avesta | 26 | 6 | 3 | 1 | 116.3 | 115.9 | 3.9 |
| Ge_ez | 26 | 6 | 1 | 3 | 116.4 | 112.8 | 4.2 |
| Glagolitic | 45 | 0 | 9 | 5 | 140.4 | 141.2 | 4.6 |
| Gurmukhi | 45 | 2 | 14 | 7 | 143.1 | 144.8 | 5.1 |
| Kannada | 41 | 3 | 11 | 5 | 137.4 | 138.8 | 4.5 |
| Keble | 26 | 2 | 3 | 2 | 114.4 | 112.0 | 3.1 |
| Malayalam | 47 | 0 | 12 | 8 | 146.6 | 148.3 | 4.5 |
| Manipuri | 40 | 1 | 7 | 8 | 134.9 | 135.7 | 4.3 |
| Mongolian | 30 | 6 | 1 | 0 | 122.0 | 119.6 | 4.1 |
| Old Church S. | 45 | 0 | 7 | 3 | 140.1 | 142.4 | 4.9 |
| Oriya | 46 | 3 | 14 | 9 | 144.6 | 143.9 | 5.4 |
| Sylheti | 28 | 22 | 2 | 5 | 117.3 | 117.2 | 4.8 |
| Syriac_Serto | 23 | 15 | 1 | 4 | 112.5 | 118.0 | 4.1 |
| Tengwar | 25 | 16 | 1 | 4 | 113.0 | 110.6 | 4.7 |
| Tibetan | 42 | 0 | 8 | 3 | 139.9 | 135.7 | 5.1 |
| ULOG | 26 | 14 | 1 | 6 | 114.5 | 110.8 | 4.2 |
| Average | | 6.4 | 5.1 | **4.6** | 129.3 | 124.5 | **4.3** |

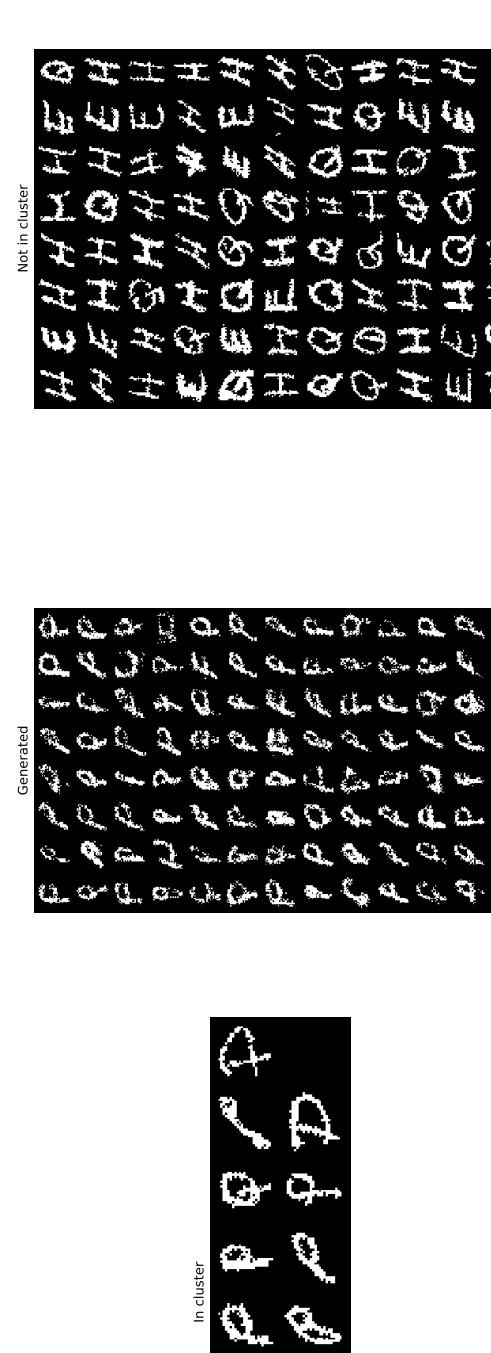

Figure 6: A step of filtering for 100 EMNIST images with 4 clusters. (Left) images belong to the cluster identified by the filtering step. (Middle) Images generated by decoding random latent vectors $z \sim p(z)$ passed through the decoder, with cluster context vector extracted from the filtering step. (Right) Images *do not* belong to the cluster identified the filtering step.

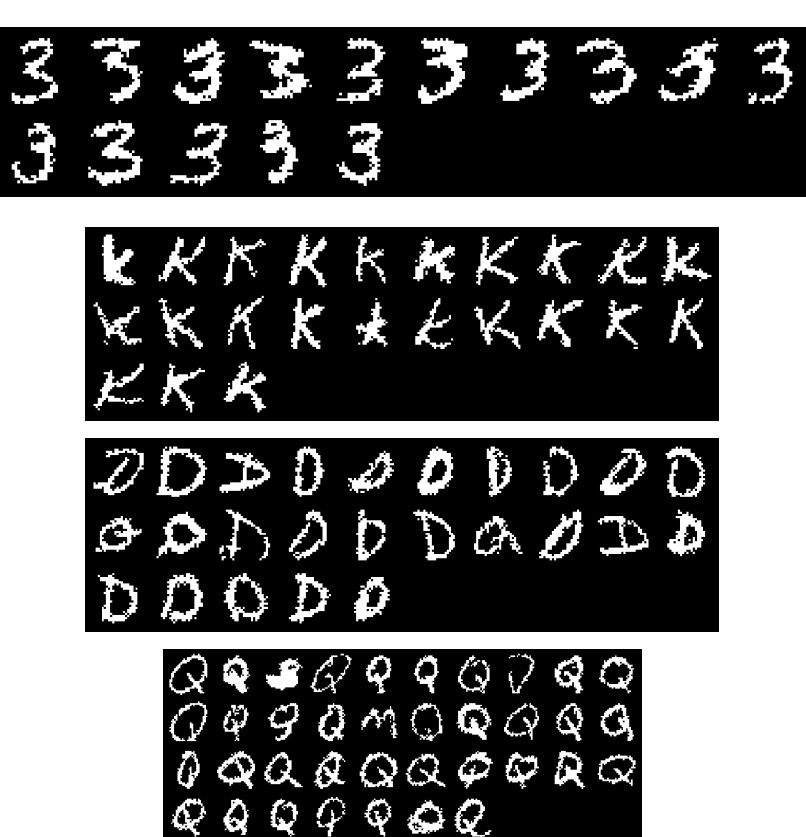

Figure 7: Clustering results of 100 EMNIST images with 4 ground-truth clusters by iterative filtering. Each block corresponds to a cluster.

