# OpenReview forum: "Deep amortized clustering"
_ICLR.cc/2020/Conference — Reject_

### Official Review · AnonReviewer3 · 2019-10-11
**Official Blind Review #3**

**Rating:** 3

**Review:**

This paper proposed a deep amortized clustering framework which learns to cluster data efficiently  based on the combination of set transformer and amortized clustering.    The main motivation is to learn clustering rules from labeled data sets and generalize to new data sets, so as to avoid manually defining clustering criterion.

Not an expert in this domain, I feel that the paper is not easy to read and many intuitions readers might be interested in are not explained very well. For example, the authors mentioned that the method proceeds sequentially, and each step identifies one cluster (the easiest cluster). However, in practical situations clusters might be meaningful only when considered in a global context, and / or under a certain scale, and more discussions are needed on how the proposed method achieves these goals.  Also, how the set transformer extracts  information useful for clustering is very unclear and needs more elaborations.

The authors used anchor points in harder problems, where the anchor points are uniformly sampled from the input data. One concern is that random sampling may lead to fluctuations in the learning process as well as very close anchor points which can be harmful for clustering.

The visualization of identified clusters seems a bit misleading. Some very compact clusters seem to be split into halfs (or with fragments of different colors) and does this indicate failed clustering on these simple data sets?

Finally, whether useful rules can be learned for clustering from labeled data is still quite open and authors may want to give some convincing examples of such rules'' for which existing clustering criterion will fail but with learned rules it can be resolved. It looks to be that the result has to do with the clustering structures of the labeled data and how can one be sure that the training data have a similar clustering structure with the to-be-clustered-data? Without answering this basic concerns, the proposed method may be hard to be accepted.

**Experience Assessment:**

I do not know much about this area.

**Review Assessment: Checking Correctness Of Derivations And Theory:**

I did not assess the derivations or theory.

**Review Assessment: Checking Correctness Of Experiments:**

I assessed the sensibility of the experiments.

**Review Assessment: Thoroughness In Paper Reading:**

I read the paper at least twice and used my best judgement in assessing the paper.

---

> ### Author Response · Authors · 2019-11-08
> **Response to the review**
>
> [Clusters only meaningful in global context] Please note that our network takes the entire set as an input for each step of filtering. Our system filters out a single cluster at a time, taking into account the topology of the whole dataset. Although we cannot describe what is going on under the hood of DAC (interpreting any neural network is a very challenging problem), our intuition is that its self-attention layers learn to identify clusters by comparing nearby data points while also taking global context into account. Rather than interpreting how our model extracts cluster information, we demonstrated through experiments that its learned strategy can successfully cluster previously unseen datasets in many different settings.
>
> [Anchor points] We agree with your point that bad anchor points may harm clustering, in the case of synthetic 2D data. Empirically, we found that the anchor point method is very effective in high-dimensional image datasets; in the paper, we apply anchored filtering only to image data. Our intuition for this is that since the anchor method selects a cluster for the model, it accelerates training in the early stages where the model must learn low-level features from scratch. Additionally, anchoring forces the network to not be biased towards always identifying a specific cluster (e.g. leftmost, darkest…).
>
> [Visualization] Please refer to our response to R2.
>
> [Are the learned clustering rules useful?] Think of the warped Gaussian datasets (sec 5.2) for instance. If we naively apply MoG in this case, the algorithm would completely fail because of the wrong assumption of the cluster shapes. On the other hand, DAC makes no assumption on the shapes of the clusters. DAC succeeded in learning the densities of clusters along with a strategy for assigning data points to those clusters, based on only the partition structures of the training datasets. I think this demonstrates that our argument is valid.
> As we clearly stated, our algorithm assumes that the datasets to be clustered are similar to the datasets seen during training. The model will fail If this assumption does not hold. However, please note that any machine learning algorithm based on empirical risk minimization would fail in this case. Adapting to change in dataset distribution is, in general, a very challenging open question and is beyond the scope of our paper.

---

> > ### Comment · AnonReviewer3 · 2019-11-13
> > **My Concerns**
> >
> > The fragmented clusters look really bad considering that these clusters themselves are so well separated but still authors argue that splitting these tight clusters should not be called failure. I do not agree with this claim.
> >
> > Traditional learning algorithm, of course, is all based on the assumptuion that raining and testing data have similar distribtuion. This, however, can be reasonably achieved;
> >
> > In the proposed paper, it is much harder to practically guarantee that the training and testing data have the same clustering structure. Note that not only clustering structure similarity can be more difficult to handle than ``distribution similarity'', but in practice without clustering the testing data , how can you make sure the training data has a similar clustering structure? this is a chicken-and-egg problem which is not feasible

---

### Official Review · AnonReviewer2 · 2019-10-25
**Official Blind Review #2**

**Rating:** 3

**Review:**

Summary:
The paper presents an amortized clustering method, called DAC, which is a neural architecture that allows efficient data clustering using a few forward passes. The proposed method is essentially based on the idea behind set-input neural networks [1], which consists of modeling the interaction between instances within a given dataset. Compared with the previous work [1], the main difference is that DAC does not need to specify the number of clusters, as in the case of Bayesian nonparametrics, making it more flexible for clustering complex datasets. It is empirically shown that DAC can efficiently and accurately cluster new datasets coming from the same distribution for both synthetic and image data.

Strengths:
Overall, I think the paper is well written and the relationship to previous works is well described. The empirical results seem promising, especially in terms of computational efficiency. The authors conduct some experiments on relatively large datasets, such as miniImageNet and tiereImageNet, which is indeed crucial for the practical applications of the proposed model.

Weaknesses:
- I think this is a good paper, but my major concern is the limited theoretical contribution, given the fact that this work is mainly based on set-input neural networks introduced in Ref. [1]. I would like the authors to clarify a bit more the novelty of the paper.
- The authors claim that DAC can process data points in parallel while Ref. [2] uses a sequential sampling procedure. However, there does not seem to be sufficient details on how to parallelize the proposed algorithm.
- As shown in Figs. 1 and 4, it seems that some clusters are split into two or three fragments. I think this simply means the failure of the proposed method on synthetic data.
- As also mentioned in the discussion on page 8, it would be important to consider uncertainties in cluster assignments, as already done in Ref. [2]. I would recommend the authors to provide some insight on how to take the cluster assignment uncertainty into account within current model.

At the moment, I recommend a weak reject as the technical contribution of the paper seems rather limited, but I could be open to increasing my score if my concerns are addressed.

References:
[1] J. Lee, Y. Lee, J. Kim, A. R. Kosiorek, S. Choi, and Y. W. Teh. Set transformer: a framework for attention-based permutation-invariant neural networks. In Proceedings of International Conference on Machine Learning, 2019.
[2] A. Pakman, Y. Wang, C. Mitelut, J. Lee, and L. Paninski. Discrete neural processes. ArXiv:1901.00409, 2019.

**Experience Assessment:**

I have published one or two papers in this area.

**Review Assessment: Checking Correctness Of Derivations And Theory:**

I assessed the sensibility of the derivations and theory.

**Review Assessment: Checking Correctness Of Experiments:**

I assessed the sensibility of the experiments.

**Review Assessment: Thoroughness In Paper Reading:**

I read the paper at least twice and used my best judgement in assessing the paper.

---

> ### Author Response · Authors · 2019-11-08
> **Response to the review**
>
> [Novelty] We argue that our paper is not a mere application of the Set Transformer proposed in Lee et al., 2019. The core contribution of this paper is a framework for training a flexible and robust amortized clustering model. The Set Transformer is simply the architectural backbone we used to implement our framework, much like how one might use a standard ResNet backbone to verify a new idea in computer vision. We summarize our contribution as follows:
> 1. A new learning framework that decomposes the clustering problem into a sequence of filtering problems along with a learning objective to achieve this. This is completely different from the amortized clustering objective used in Lee et al., 2019, where they try to maximize the overall likelihood of datasets.
> 2. A way to apply the amortized clustering system beyond simple parametric families. We illustrated two examples (mixture of MAFs and mixture of neural statisticians) and demonstrated their effectiveness. In contrast, the clustering model of (Lee et al., 2019) is restricted to a mixture of Gaussians model and thus cannot be used on e.g. miniImagenet.
>
> [Parallel processing] The parallelization of our model is fairly simple, and our provided code is implemented in this way. Assume we are given D datasets, each with N images of size (H, W, C). The input to our DAC model would be a 5-dimensional tensor with shape [D, N, H, W, C]. After each step of filtering, we simply apply the appropriate filtering mask to each of the D datasets and feed the resulting 5-dimensional tensor through the network again for the next step. This whole process is easily parallelized on modern deep learning libraries (we used PyTorch). In contrast, the neural clustering process requires sequential sampling that iterates over datapoints, which cannot be parallelized. The number of forward passes required for each dataset is O(N) for the Neural Clustering Process (NCP) and O(K) for DAC. This allowed us to consider tasks with up to N=3000 2D points or N=100 images, while such large problems would be very time-consuming for NCP.
>
> [Fragmented clusters] While we agree that DAC did not find the “perfect” clustering on those examples, we think it is rather harsh to say the algorithm failed because of the fragmented cluster. Such fragmented clusters are commonly found in MoG-based clustering methods, due to the diagonal covariance assumption and the structure of the loss. We further point out that the DAC models used to generate Figs 1, 4 were trained on datasets with at most 4 clusters and 1000 data points, and were tested on a dataset with far more clusters and data points. Figs 1, 4 demonstrate that DAC is capable of generalizing to such drastically different datasets, which we attribute to our iterative filtering procedure. Results in Table 1 show that DAC was significantly better than previous methods in this particular task.
>
> [Uncertainty in cluster assignments] A simple way to accomplish this in DAC, which we considered in early experiments, is to use the assignment probability (instead of hard assignment masks) as input to the next filtering step. We excluded this from the manuscript as it did not increase clustering accuracy. As stated on page 8, we believe such uncertainty-aware amortized clustering is an important and interesting research topic, and we plan to investigate this problem in more detail in future work.

---

### Official Review · AnonReviewer1 · 2019-10-26
**Official Blind Review #1**

**Rating:** 3

**Review:**

[Overview]

In this paper, the authors proposed a new clustering method called deep amortized clustering (DAC). Inspired by Lee et al 2019, the authors exploited a transformer to gather the contextual information across different dataset points and then predict the cluster label for each data point. The main difference from Lee et al is that the proposed DAC sequentially estimate the cluster labels for the data points and thus more flexible to estimate the number of clusters in the whole dataset. Based on the proposed DAC method, the authors evaluated the performance on both unsupervised clustering and supervised clustering tasks. it turns out the proposed method has achieved better or comparable performance to previous work on various datasets while hold less computational cost.

[Pros]

1. In this paper, the authors extended the clustering method in Lee et al to a new method called Deep Amortized Clustering (DAC) for data clustering. In this new method, the number of clusters can be unknown at the beginning and the model itself will sequentially cluster the data points into different groups until are data points have been assigned to some clusters. This is an interesting method in that it does not need to specify the number of clusters at the beginning, and thus become more flexible.

2. To achieve the DAC, the authors proposed two losses, one is Minimum Loss Filtering (MLF) and one is Anchor Filtering to cope with either multi-gaussian-distributed data points or even harder datasets. Meanwhile, the authors also proposed to estimate the density P(x; \theta) in the case that the distribution is not knowing in prior.

3. The authors evaluated the proposed method on both synthetic dataset and realistic dataset. On the synthetic dataset, the proposed method is compared with VBDPM and ACT-ST, two methods that can cope with dataset with unknown number of clusters. On the realistic datasets, the authors evaluated on the EMNIST which is of non-MoG distribution. Besides, the authors further evaluated the method on MiniImageNet features and Omniglot dataset, and showcased comparable performance to previous methods but much shorter running  time.

[Cons]

1. Overall, the paper is poorly written and organized. First, the notations in the method section are hard to follow. There are a number of notations which are all capital characters, either representing a function or a method. Second, the whole training process and inference process of the proposed method is not clear to me. How the model is trained on the training set, what are the learnable parameters in the proposed model and what are the settings for the hyper-parameters, etc. Third, it is hard to get the takeaway messages from the experiment sections. The experimental settings for each subsection are not very clearly explained, and the analysis on the experimental results are also vague.

2. In the method, the authors proposed Minimum Loss Filtering (MLF) for clustering with the loss function in Eq(5). During training, the authors use some training data with ground-truth labels to optimize the loss function. However, it is not clear which parameters will be learned in the optimization. Also, after the training, what the exact inference procedure should be is also not clear to me. Overall, it is really hard to me to follow this section on the filtering process. The authors should definitely describe the process more clearly.

3. The experimental results shown in the paper are hard to interpret. First, the setting for each experiment is not clear to me. In Figure 3, it is hard to understand the figures clearly. In table 2 and table 3, some of the clustering methods are deterministic, such as K-means, Spectral clustering, DEC. However, some other clustering methods are learning to clustering methods, such as KCL and MCL. Putting all of the numbers in the same table is confusing and make it hard to compare. I would suggest the authors make a clear distinction between different methods: 1) deep clustering methods which directly cluster on top of the test set; 2) learning to clustering method which learn some parameters on training set and then generalize to test set; 3) amortized clustering, which also learn some parameters on training and then test on the test set with just one forward pass. Splitting the testing results into three group will be helpful to the readers to understand the paper and the proposed method.

4. Another missed part in the model is the ablation study. How sensitive the model is to different training set, e.g., different training dataset size, different number of training clusters, and different hyper-parameters, etc. Without these information, it is hard to know how well robust the proposed DAC method can generalize.

5. Finally, the proposed method was built upon Lee at al 2019, to extend the previous method to a sequential clustering problem. I think a title "deep amortized clustering" is a bit misleading and exaggerated on the proposed method.

[Summary]

In this paper, the authors proposed a new method called Deep Amortized Clustering (DAC) for amortized clustering. Unlike the previous work Lee et al, the proposed DAC sequentially filter the data points from the whole set and construct the clusters gradually. This is a meaningful method in that it can be applied to those data without explicit number of clusters. However, the presentation of the method and experiments make it hard to follow, and thus hard to capture the contributions of the proposed method, and also its capacity. As also mentioned above, I would highly suggest the authors revise the paper so that it can present better the method and the experimental section.

**Experience Assessment:**

I have published one or two papers in this area.

**Review Assessment: Checking Correctness Of Derivations And Theory:**

I assessed the sensibility of the derivations and theory.

**Review Assessment: Checking Correctness Of Experiments:**

I carefully checked the experiments.

**Review Assessment: Thoroughness In Paper Reading:**

I read the paper at least twice and used my best judgement in assessing the paper.

---

> ### Author Response · Authors · 2019-11-08
> **Response to the review**
>
> 1. The main takeaway from our experiment is, 1) DAC trained in the way described in the paper can actually “amortize” the clustering procedure so that it can accurately and efficiently cluster new datasets. 2) DAC can adapt to different numbers of clusters in datasets and can infer various cluster distributions. 3) Even though DAC is an amortized method, its clustering accuracy is comparable to or sometimes better than other clustering methods trained on datasets from scratch.
>
> 2. The parameters to be learned are the parameters of the set-input neural networks (ISAB, PMA, MABs, …). We will make this clearer in the paper. The inference procedure after the training is the following. We simply feed the entire dataset into the network, which outputs \theta and. \theta and m are cluster parameters and mask, respectively (see eq 6). We repeat this procedure until either all data points are selected by a mask or m does not select any of the remaining points.
>
> 3. Clusters are color-coded along with the density contour plots. In Figure 3, each datapoint is colored according to which cluster it was assigned to, and the contour plot depicts the density of each cluster. We appreciate the suggestion to separate the table into groups of methods.
>
> 4. We showed in fig 1, 3 and Table 1 that DAC can generalize to datasets having different sizes and numbers of clusters. We are not sure whether DAC would generalize when the hyperparameters of cluster generating distribution changes, but such a setup violates the core assumption underlying our method, which is that the train- and test- datasets are generated through the same process.
>
> 5. Thanks for your suggestion; we will think about a title that is proportional to the contribution of our method.

---

### Author Response · Authors · 2019-11-08
**Overall response to the reviews**

Overall, we acknowledge that our explanation of our method along with our choice of notation could confuse readers, especially those who are not familiar with this line of research. We will improve the overall presentation of our manuscript to be more clear and self-contained. Please refer to the individual comments below.

---

### Decision · Program_Chairs · 2019-12-19

**Decision:**

Reject

**Comment:**

This paper introduces a new clustering method, which builds upon the work introduced by Lee et al, 2019 - contextual information across different dataset samples is gathered with a transformer, and then used to predict the cluster label for a given sample. All reviewers agree the writing should be improved and clarified. The novelty is also on the low side, given the previous work by Lee et al. Experiments should be more convincing.